# INVARIANT LEARNING WITH PARTIAL GROUP LABELS

## ABSTRACT

Learning invariant representations is an important requirement when training machine learning models that are driven by spurious correlations in the datasets. These spurious correlations, between input samples and the target labels, wrongly direct the neural network predictions resulting in poor performance on certain groups, especially the minority groups. Robust training against these spurious correlations requires the knowledge of group membership for every sample. Such a requirement is impractical in situations where the data labelling efforts for minority or rare groups is significantly laborious or where the individuals comprising the dataset choose to conceal sensitive information. On the other hand, the presence of such data collection efforts result in datasets that contain partially labelled group information. Recent works have tackled the fully unsupervised scenario where no labels for groups are available. Thus, we aim to fill the missing gap in the literature by tackling a more realistic setting that can leverage partially available sensitive or group information during training. First, we construct a constraint set and derive a high probability bound for the group assignment to belong to the set. Second, we propose an algorithm that optimizes for the worst-off group assignments from the constraint set. Through experiments on image and tabular datasets, we show improvements in the minority group's performance while preserving overall aggregate accuracy across groups.

## 1 INTRODUCTION

Neural networks being overly biased to certain groups of the data is an increasing concern within the machine learning community [1]. A primary cause for bias against specific groups is the presence of extraneous attributes in the datasets that wrongly direct the model responses [38]. Such extraneous attributes are features that need to be controlled for. For example, in computer vision tasks such as image classification or object detection, an extraneous attribute could correspond to the background in an image or a co-occurring object irrelevant to the task, e.g. a person making a speech in a football field could be predicted as playing football [6]. The presence of such extraneous attributes warrant a model to derive the predictions by making spurious correlations to extraneous features in an image rather than an actual object of interest. An inevitable consequence of such correlations to extraneous attributes is disparities in performance across different groups within the dataset. Specifically, if certain groups form a minority, a model can simply *cheat* by having a high overall aggregate accuracy but poor minority group accuracy [24].

Existing works for this problem [2, 30] operate in the regime where the number of groups, likely to be adversely impacted through spurious correlations, are known apriori. Further, they assume a complete knowledge of the group membership of individual samples in the training dataset. While these methods have been proven effective, it is not realistic to assume access to the group labels for every sample. Consider the scenario where the minority / majority groups could be defined by demographic information such as *gender* or *race*. An individual can simply choose not to reveal this information due to privacy considerations [16]. Alternatively, in medical image settings a label class could contain unrecognized subgroups that demand significant burden on the data labelling efforts [33]. An example of such unrecognized subgroups could be a lung cancer detection problem, where the class label could comprise of groups such as solid/subsolid tumors and central/peripheral neoplasms [24], with many of these groups naturally forming a minority in the dataset. In this work, we consider a setting where a significant portion of the training data is devoid of group labels. We choose to fill a missing gap in the literature where several works bifurcate into methods that either are fully supervised or fully unsupervised in terms of the groups labels. Knowledge on the number

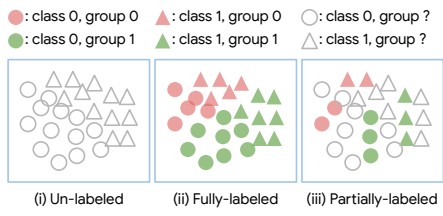 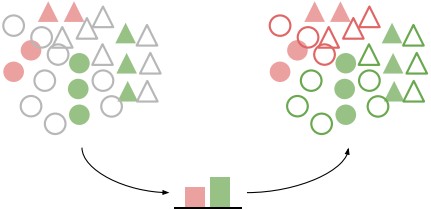

(a) Problem settings.

(b) Find the worst-case group assignment with the marginal distribution constraint.

Figure 1: (a) We introduce Worst-off DRO, an invariant learning algorithm for partial group labeled data (as shown in (iii)). This is in contrast with other settings, such as (unsupervised) DRO [14] where (i) no group labels are available at train time, or Group DRO [30] when (ii) group labels for all training examples are required. (b) Worst-off DRO finds the worst-case group assignment for missing data with a marginal distribution constraint, which may be given as side information, or estimated from labeled counterparts, and optimizes similarly to that of Group DRO. The constraint set defined by the marginal distribution includes the ground-truth group labels with high probability, ensuring the training objective of Worst-off DRO to be an upper bound of that of Group DRO with ground-truth group labels of entire data.

of groups in the data makes it convenient to obtain group labels for a tiny portion of the data or take advantage of an existing labelled samples if available. Hence, we address the following research question: *Can we train a model that is invariant to group membership using partially labelled data?*

We answer the question using a framework of distributionally robust optimization (DRO) [23, 32]. DRO allows for a training routine that optimizes for the worst-case training loss over predefined set of groups closely connected to the Rawlsian fairness measure [27]. When the group membership is fully known, the method simply upweights/downweights average training loss of different groups through the course of training [30]. The application of DRO to the partial group label setting poses significant challenges: (1) the lack of group label makes it infeasible to compute the worst-off group loss; and (2) optimizing only for the high-loss samples, by considering them as a worst-off group, discards considerable portion of the training data adversely impacting the overall accuracy.

In light of these challenges, we make the following **contributions** in this paper. We construct a constraint set that can with high probability encompass the group labels of the unlabeled portion of the data. We optimize for the a worst-off soft group assignment from the constraint set which upper bounds the DRO objective with true group membership. Approximately, high loss samples are assigned to groups with a higher weight and lower marginal probability improving the worst-off group accuracies. Further, the low loss samples are not discarded thus preserving the aggregate accuracy across groups. We show experiments on two images datasets and one tabular datasets and contrast the performance against several baselines.

## 2 RELATED WORK

Distributionally robust optimization [4, 11, 23] has been studied as a way to train robust ML models across multiple groups and environments.

**Group Robust Optimization.** When group information is known at train time, Group DRO [15, 22, 25, 30, 40] or Invariant Risk Minimization (IRM) [2] could be employed to improve the performance over multiple groups. Specifically, Group DRO proceeds by minimizing the loss of the group with the largest loss, while IRM enforces a shared predictor across multiple environments to be optimal in the form of a Lagrangian multiplier.

**Robust Optimization without Demographics.** As the group information may not be always available reliably, several studies have been focusing on developing methods that remove or reduce their dependence on the group information. [14] has developed a method based on the distributionally robust optimization that minimizes the loss of the samples with losses larger than a certain threshold. [18] has proposed to reweight the samples in an adversarial way so that the high loss sample could receive more attention over the course of training. Moreover, [20] has proposed a simple yet effective two-stage approach called Just-Train-Twice that trains a model by upweighting samples with high losses from the initial ERM model.

While above methods are based on optimizing high loss samples based on sample reweighting, GEORGE [33] and EIIL [7] are two-stage optimization methods, where in the first stage the group

or environment labels are inferred and in the second stage Group DRO or IRM optimization are employed with inferred group labels from the first stage. Our method shares a similar idea to GEORGE or EIIL in that we infer group labels for unlabeled data and apply a (variant of) Group DRO for optimization. However, we infer group labels in a way that the objective becomes the lower bound of that of Group DRO with true group labels with high probability. Such theoretical guarantee is important for safety-critical ML applications.

# 3 METHODOLOGY

We introduce our robust optimization framework, Worst-off DRO, with partial group information. We revisit the GroupDRO in Section 3.1 and detail our method in Section 3.2 and 3.3. In Section 3.4, we describe a practical method for optimization.

## 3.1 PRELIMINARY: GROUPDRO

Let $x \in \mathcal{X} \subset \mathbb{R}^d$ be data descriptors, $y \in \mathcal{Y} \subset \{0, 1\}$ be target labels, and $g \in \mathcal{G} \subset \{1, ..., M\}$ be group labels. We assume training a neural network parameterized by the weights $w$ that corresponds to a per-sample loss $l(x, y; w)$. Given data triplets $\{(x_i, y_i, g_i)\}_{i=1}^N$, we seek to optimize $w$ for the Rawlsian criterion [14, 27, 39], which minimizes the loss of the worst-off group, as follows:

$$\min_w \max_{g \in \mathcal{G}} \mathbb{E}\big[l(x, y; w)|g\big]. \tag{1}$$

[30] proposed a practical algorithm to optimize Equation 1, called *Group DRO*. This method optimizes a weighted expected loss across all groups. These weights over the groups, denoted by $q$, are drawn from a simplex $\Delta^M$. The objective function is as described below,

$$\mathcal{L}_{\text{GDRO}} = \min_w \max_{q \in \Delta^M} \sum_{j=1}^M q_j \Big[ \frac{\sum_{i=1}^N \mathbb{1}\{g_i = j\} l(x_i, y_i, w)}{\sum_{i=1}^N \mathbb{1}\{g_i = j\}} \Big] \tag{2}$$

## 3.2 WORST-OFF DRO

In this work, we are interested in training a distributionally robust neural network when group labels are only partially available in the entire dataset. That is, our training dataset constitutes of the fully-labeled dataset $\{(x_i, y_i, g_i^\star)\}_{i=1}^K$ and the task-labeled dataset $\{(x_i, y_i, -)\}_{i=K+1}^N$, where $-$ indicates the missing group labels.

As noted in Equation 2, the Group DRO requires group labels of entire dataset. When some of them are missing, we propose to optimize for the following objective::

$$\mathcal{L}_{\text{WDRO}}(\mathcal{C}) = \min_w \max_{q \in \Delta^M} \max_{\{\hat{g}\} \in \mathcal{C}} \sum_{j=1}^M q_j \Big[ \frac{\sum_{i=1}^N \mathbb{1}\{\hat{g}_i = j\} l(x_i, y_i, w)}{\sum_{i=1}^N \mathbb{1}\{\hat{g}_i = j\}} \Big] \tag{3}$$

where $\mathcal{C}$ is a set of group assignments $\{\hat{g}_i\}_{i=1}^N$ satisfying $\hat{g}_i = g_i^\star, \forall i \leq K$. We call the objective in Equation 3 a **Worst-off DRO** as it optimizes neural network parameters with respect to the worst-off group assignment in some constraint set $\mathcal{C}$.

Observe that the Worst-off DRO objective forms an upper bound to the Group DRO objective when evaluated at the ground-truth group labels. This is rather a straightforward consequence from the fact that the ground-truth group assignment $\{g_i^\star\}_{i=1}^N$ falls within the constraint set $\mathcal{C}$. However, for safety-critical applications, such as learning a fair classifier, it is important that the optimal objective (i.e., Group DRO with a ground-truth group assignment) is bounded by the learning objective used in optimization. This is simply because optimizing the proposed learning objective **guarantees** that the corresponding lower bound ground-truth Group DRO is also optimized (assuming same parameters for both the methods). Conversely, objectives of methods such as EIIL [7] or GEORGE [33], which optimize the Group DRO or IRM objectives using pseudo group labels, would be difficult to compare with $\mathcal{L}_{\text{GDRO}}$ as it depends on a heuristic to obtain a single set of pseudo group labels.

## 3.3 REDUCING CONSTRAINT SET WITH MARGINAL DISTRIBUTION CONSTRAINT

It is clear that the constraint set $\mathcal{C}$ plays an important role that connects Worst-off DRO to Group DRO. Specifically, the Worst-off DRO objective can be made a tighter bound to that of Group DRO

by further constraining $\mathcal{C}$ so long as it contains the ground-truth group assignment $\{g_i^\star\}_{i=1}^N$. In the subsequent paragraph, we describe how we reduce the constraint set while retaining the ground-truth group assignment using a marginal distribution constraint. These constraints may be given as a side information or could be estimated from the small set of partial group labels under certain conditions.

Let $\mathcal{C}_{\mathbf{p},\epsilon} \subset \mathcal{C}$ is a subset of $\mathcal{C}$ whose elements $\{g_i\}_{i=1}^N$ satisfy the following condition:

$$g_i = g_i^\star, \forall i \le K, \tag{4}$$

$$|\frac{1}{N} \sum_{i=1}^N \mathbb{1}\{g_i = j\} - p_j| \le \epsilon, \forall j \le M, \tag{5}$$

where Equation 4 implies that the true group labels are assigned whenever available, and Equation 5 implies that the data marginal distribution should be close to the marginal distribution $\mathbf{p}$. Then, for any marginal distribution $\mathbf{p}$ and $\epsilon > 0$, it is easy to show $\mathcal{L}_{\text{WDRO}}(\mathcal{C}_{\mathbf{p},\epsilon}) \le \mathcal{L}_{\text{WDRO}}(\mathcal{C})$ as $\mathcal{C}_{\mathbf{p},\epsilon} \subset \mathcal{C}$. Moreover, we will see in Lemma 1 that, with high probability, the constraint set $\mathcal{C}_{\mathbf{p}^\star,\epsilon}$ with the true marginal distribution $\mathbf{p}^\star$ contains the true group assignment $\{g^\star\}$.

**Lemma 1.** *The constraint set $\mathcal{C}_{\mathbf{p}^\star,\epsilon}$ contains the true group labels $\{g_i^\star\}_{i=1}^N$ with high probability:*

$$P(\{g_i^\star\}_{i=1}^N \in \mathcal{C}_{\mathbf{p}^\star,\epsilon}) \ge 1 - 2e^{-2N\epsilon^2} \tag{6}$$

The proof is in Appendix A.1. As in Equation 6, the probability of the constraint set containing the true group labels gets closer to 1 by allowing a larger variance ($\epsilon$) from the true marginal distribution. For fixed $\epsilon > 0$, the probability gets closer to 1 as we increase the number of *unlabeled* data ($N$).

Finally, this implies that $\mathcal{L}_{\text{WDRO}}(\mathcal{C}_{\mathbf{p}^\star,\epsilon})$ is an upper bound to that of Group DRO:

$$\mathcal{L}_{\text{GDRO}} \underset{\text{w.h.p}}{\le} \mathcal{L}_{\text{WDRO}}(\mathcal{C}_{\mathbf{p}^\star,\epsilon}) \le \mathcal{L}_{\text{WDRO}}(\mathcal{C})$$

In practice, however, the true marginal distribution $\mathbf{p}^\star$ may not be available. For our setting where group labels are partially available, with an assumption that group labels are missing completely at random (MCAR) [29], the true marginal distribution could be estimated from the subset of data with group labels. This again allows us to formulate a constraint set that contains the ground-truth group assignment with high probability.

To be more specific, let $\bar{\mathbf{p}}$ be the estimate of the marginal distribution from $\{(x_i, y_i, g_i^\star)\}_{i=1}^K$.

**Lemma 2.** *The constraint set $\mathcal{C}_{\bar{\mathbf{p}},\delta+\epsilon}$ contains the true group labels $\{g_i^\star\}_{i=1}^N$ with high probability:*

$$P(\{g_i^\star\}_{i=1}^N \in \mathcal{C}_{\bar{\mathbf{p}},\delta+\epsilon}) \ge 1 - 2e^{-2N\epsilon^2} - 2e^{-2K\delta^2} \tag{7}$$

We provide a proof in Appendix A.1. Here, $\delta$ is introduced to take into account the estimation error of the true marginal distribution $\mathbf{p}^\star$. When $K$, the number of labeled data, is large, the bound in Equation 7 is close to 1.

$$\text{Assuming MCAR, for large } K: \mathcal{L}_{\text{GDRO}} \underset{\text{w.h.p}}{\le} \mathcal{L}_{\text{WDRO}}(\mathcal{C}_{\bar{\mathbf{p}},\delta+\epsilon})$$

## 3.4 A PRACTICAL OPTIMIZATION ALGORITHM

We are interested in solving the optimization problem $\mathcal{L}_{\text{WDRO}}(\mathcal{C}_{\mathbf{p},\epsilon})$. Unfortunately, the inner maximization problem with respect to the group assignments $\{\hat{g}\}$ in Equation 3 is challenging as variables are discrete and the objective cannot be decomposed due to the marginal distribution constraint. In this section, we describe an optimization recipe with a few approximations.

First, we propose to use a soft group assignments. This not only converts the problem into continuous optimization problem, but also accommodates inherent uncertainties in group assignment

---

**Algorithm 1** Worst-off DRO Algorithm

---

1: *Input:* Fully-labelled dataset $\{(x_i, y_i, g_i^\star)\}_{i=1}^K$, task-labeled dataset $\{(x_i, y_i, -)\}_{i=K+1}^N$
2: *Initialization:* learning rates $\eta_w$ and $\eta_q$, Marginal distribution $\bar{\mathbf{p}}$, $\epsilon$
3: *Parameters:* Group Weights $q_j$, Worst-off DRO group assignments $\hat{g}$,
   Neural network parameter $w$
4: **for** $t = 0, 1, 2, ..., T$ **do**
5:     $\{\hat{g}^t\} \leftarrow \max_{\{\hat{g}\} \in \mathcal{C}_{\bar{\mathbf{p}}, \epsilon}} \sum_{j=1}^M q_j^t \frac{\sum_i \hat{g}_{ij} l(x_i, y_i; w^t)}{\sum_i \hat{g}_{ij}}$ where, $\mathcal{C}_{\bar{\mathbf{p}}, \epsilon}$ as defined in Equation 9.
6:     Gradient descent on $w$:
       $w^{t+1} \leftarrow w^t - \eta_w \nabla_w \sum_{j=1}^M q_j^t \frac{\sum_i \hat{g}_{ij}^t l(x_i, y_i; w)}{\sum_i \hat{g}_{ij}^t}$
7:     Exponential ascent on $q$:
       $q^{t+1} \leftarrow q^t \exp \left( \eta_q \nabla_q \sum_{j=1}^M q_j \frac{\sum_i \hat{g}_{ij}^t l(x_i, y_i; w^{t+1})}{\sum_i \hat{g}_{ij}^t} \right)$
8: **end for**
9: *Output:* Trained neural network parameters $w^T$

---

for data with missing group labels. Specifically, for each data, we retain a soft group assignment $\hat{g}_i \in \Delta^M$, and optimize the Worst-off DRO objective over the constraint set $\mathcal{C}_{\bar{p}, \epsilon}$ as defined below:

$$\min_w \max_{q \in \Delta^M} \max_{\{\hat{g}\} \in \mathcal{C}_{\mathbf{p}, \epsilon}} \sum_{j=1}^M q_j \left[ \frac{\sum_{i=1}^N \hat{g}_{ij} l(x_i, y_i, w)}{\sum_{i=1}^N \hat{g}_{ij}} \right] \tag{8}$$

where the constraint set is defined as:

$$\mathcal{C}_{\bar{\mathbf{p}}, \epsilon} = \left\{ \{\hat{g}_i\}_{i=1}^N \middle| \begin{array}{l} \hat{g}_i \in \Delta^M, \forall i \leq N, \\ \hat{g}_{i(g_i^\star)} = 1, \forall i \leq K, \\ |\frac{1}{N} \sum_{i=1}^N \hat{g}_{ij} - \bar{\mathbf{p}}_j| \leq \epsilon, \forall j \leq M \end{array} \right\} \tag{9}$$

The first condition ensures assignments in the probability simplex, second one ensures assignments are consistent with ground-truth for labeled data, and the third one ensures the data marginal distribution follows the provided distribution. This resulting optimization problem is solved using off-the-shelf CVXPY solver [8]. More details on this and an example are provided in the Appendix A.2.

Finally, we alternate optimization over $w$, $q$ and $\{\hat{g}\}$ as in Algorithm 1. That is, we first solve inner maximization over $\{\hat{g}\}$, and conduct gradient descent on $w$ and the exponential gradient ascent on $q$, and iterate. An exponential ascent on $q$ achieves smaller losses for linear predictors (like $q$) [17].

## 4 EXPERIMENTS

We test the efficacy of our method on image and tabular datasets, each of which consists of samples from mutually exclusive groups or environments. These groups are indicative of the background or an RGB identification for image datasets, and attributes such as gender or race for tabular datasets. As discussed in Section 1, one or more of the available groups form a minority in terms of the sample size and demographics. The presence of minority groups results in a possible scenario where the aggregate performance is (falsely) remarkable, because evaluations are dominated by larger groups, even though the performance on the minority groups is poor. In our experiments we assume group numbers to be known but group labels are *missing completely at random* at a fixed rate at train time.

In Section 4.1, we outline our baselines for comparison, our metrics of evaluation and the model selection strategy. In Section 4.2, we describe each dataset in detail and highlight the differences across the groups within the dataset. All the quantitative results are available in Table 2 and per-group summary statistics are present in Table 1. More analysis of our method is provided in Section 4.3.

### 4.1 EXPERIMENTAL SETTINGS

**Baselines.** We contrast the performance of our method with respect to a few well-known baselines.

1.  **ERM:** Empirical Risk Minimization that optimizes aggregate average loss over all the samples in the training dataset.

| Dataset | # Labeled | # UnLabeled | Total samples | # Groups | # Minority Samples | # Majority Samples |
|---|---|---|---|---|---|---|
| Waterbirds | 508 | 4287 | 4795 | 4 | 5 | 113 |
| CMNIST | 3983 | 35657 | 39640 | 3 | 276 | 2138 |
| Adult | 1308 | 11635 | 12943 | 4 | 63 | 823 |
| CelebA | 8030 | 154740 | 162770 | 4 | 62 | 3547 |

Table 1: **Dataset description.** We show sample counts in labelled and unlabelled training sets, as well as counts for majority and minority groups. The number of labelled samples are about $10\%$ of total samples.

2. **Unsup DRO [14]:** Samples with losses exceeding a threshold $\eta$ are considered as a group whose average loss is optimized. Since the method doesn't require group labels, it is an unsupervised algorithm.

3. **Group DRO [30]:** A method that optimizes the Rawlsian criterion by assigning simplex weights to the groups. The group labels for individual samples are assumed to be available, hence this method is fully-supervised in terms of the group label.

4. **Group DRO (Partial):** We consider another variant of Group DRO that only uses samples with group labels at train time. We call the method *Group DRO (Partial)*, to contrast with the above baseline, *Group DRO (Oracle)*.

We compare above methods with our proposal, **Worst-off DRO**. Note that our approach requires marginal probabilities as an input to the algorithm, which are computed from the training dataset in our experiments. For baselines and Worst-off DRO implementations, samples are drawn randomly for every batch ensuring an unbiased comparison to the ERM baseline. This is unlike [30] who adopt a weighted sampling procedure which could be noisy when group labels are uncertain.

**Evaluation Metrics.** We set aside a test set whose group labels are fully available. Since all of our datasets characterize a classification task, we evaluate overall accuracies and per-group accuracies in our experiments. Specifically, we highlight the accuracy of the minority group (**min**) together with the overall (**avg**) accuracy where individual samples are equally weighted regardless of their group.

**Model Selection.** Model selection plays a crucial role when distributional differences are observed in a dataset [13]. In our problem setting, individual groups may differ from each other in the joint distribution over the data and the label space, however, the testing set resembles the training set. That is, there are no out of distribution samples and the focus is to improve robustness over a predefined set of groups common to both training and testing datasets. Consequently, among the recommendations made in [13], a training domain validation set is a feasible strategy for our problem. In our algorithm, learning rate and weight decay are important hyper-parameters. Prior works [30, 31] noted that ERM fails to optimize for the minority group's performance under high regularization regime thus necessitating an alternative. Hence, we compare our methods in this regime. We also tune for algorithmic specific hyper-parameters for each baseline. These hyper-parameters are the loss threshold in Unsup DRO ($\eta_{\text{UDRO}}$) and the step size for the group weights in Group DRO ($\eta_{\text{GDRO}}$) and in Worst-off DRO ($\eta_{\text{WDRO}}$). A list of all hyper-parameter choices used in the experiments is provided in the Appendix A.3. All the numbers reported in the paper were averaged over three random seeds.

We adopt **NVP** (novel validation procedure) [9] in our experiments. In this procedure, we first search for hyper-parameters with the best overall accuracy. Then, from the top five best performing hyper-parameters, we select the model that achieves the highest minority group accuracy. Such a procedure offers robustness to hyper-parameters in the reported numbers.

## 4.2 Quantitative Results

We describe key results on four datasets, Waterbirds [30], CMNIST [2], Adult [10], and CelebA [21].

### 4.2.1 Waterbirds

The dateset, used in [30], comprises of 4795 images of birds from the CUB dataset [37] and the backgrounds taken from the Places dataset [41]. Each image in the dataset has a background of land or water. The target labels are either "landbirds" or "waterbirds". The authors in [30] create four groups with each target label and a background class considered as a group. In this dataset the groups "landbirds" on water and "waterbirds" on land form a minority. Our results in Table 2 firstly shows that the ERM method attains a small minority group accuracy of $60\%$. All the invariant learning baselines, except for Group DRO (Partial), improve the minority group's accuracy. Next, we observe that in comparison to Group DRO (Partial), our proposed Worst-off DRO improves the

|  | Waterbirds | | CMNIST | | Adult | | CelebA | |
|---|---|---|---|---|---|---|---|---|
|  | **min** | **avg** | **min** | **avg** | **min** | **avg** | **min** | **avg** |
| Group DRO (Oracle) | 83 | 92 | 50 | 75 | 82 | 88 | 80 | 94 |
| ERM | 60 | 87 | 13 | 79 | 68 | 92 | 45 | 95 |
| Unsup DRO | 65 | 88 | 10 | 80 | 68 | 92 | 39 | 96 |
| Group DRO (Partial) | 44 | 81 | 36 | 76 | 67 | 90 | 40 | 95 |
| Worst-off DRO | **65** | 89 | **39** | 77 | **71** | 91 | **49** | 95 |

Table 2: **Quantitative Results.** For baselines, we consider an ERM, Unsup DRO [14], Group DRO (Partial) for partly labelled Group DRO [30] method, Group DRO (Oracle) for the fully supervised model. Our method Worst-off DRO improves the minority group's accuracy (**min**) while maintaining a similar overall accuracy (**avg**) relative to baselines. The accuracies are computed on the test set and are an average over three random runs. The standard deviations are provided in the Appendix Table 5.

minority group's performance by a significant margin of $21\%$. Due to this improvement, the all-group accuracy also improves by $8\%$. Minority group's performance on fully-supervised method Group DRO (Oracle) is at $83\%$ accuracy with a window of $18\%$ difference from Worst-off DRO.

### 4.2.2 COLORED MNIST (CMNIST)

CMNIST, derived from an MNIST [19], is a digit recognition dataset where each image is colored either red or green. Digits $< 5/ \geq 5$ are considered as label $0/1$. We consider three groups in our experiments. In the first two groups, label $0$ images are predominantly colored red and vice versa. In the third group, which forms a minority, we switch coloring such that the label $1$ images are predominantly colored red. Specifically, for the first two groups, the color id is sampled by flipping the target label with probabilities $0.2$ and $0.1$ respectively, while the third group with probability $0.9$. Both training and testing sets contain three groups. The overall setup for generating a given group is similar to [2]. We show the results on CMNIST in Table 2. Similar to the Waterbirds dataset, Worst-off DRO improves the minority group's accuracy compared to the ERM method. Relative to Group DRO (Partial), Worst-off DRO improves the accuracy of the minority group by $4\%$ and $1\%$ in the overall accuracy. The margin between Worst-off DRO and Group DRO (Oracle) is $11\%$. Among all the baseline, Unsup DRO attains lowest minority group accuracy of $10\%$. A large trade-off between the minority group accuracy and the all-group accuracy was seen for Unsup DRO in this dataset.

### 4.2.3 ADULT DATASET

We use a semi-synthetic version of the Adult dataset [10] for this experiment. Similar to [18], we consider race and sex as the four demographic groups. The target label is income $> 50K\$$ and is treated as label $1$. Similar to the CMNIST dataset, each group has a different correlation strength to the target label. For the purposes of the experiment, we exaggerate these spurious correlations caused by group membership close to [7]. Particularly, for samples with group label as Afican-American, we undersample examples with probability $P(y = 1 \mid group) = 0.06$ whereas for the non African-American group labels, we oversample examples with probability $P(y = 1 \mid group) = 0.94$. Table 2 indicates a $5\%$ improvement in the minority group's accuracy while maintaining the similar overall accuracy of $90\%$ compared to Group DRO (Partial). The Group DRO (Oracle) method reaches an accuracy of $82\%$ for the minority group compared to Worst-off DRO which achieves $71\%$. Evidently, ERM underperforms in terms of the minority group's accuracy and attains about $68\%$ accuracy.

### 4.2.4 CELEBA DATASET

CelebA [21] is a dataset containing about 200k celebrity faces curated from the internet. There are 40 labels available in this dataset which are annotated by a group of paid adult participants [5]. Similar to [30], we aim to predict the target attribute *Blond Hair* that is spuriously correlated to the Gender attribute. Specifically, having blond hair correlates with the female attribute. The minority group in this dataset are the images with attributes (blond, male). The proportion of samples in the minority and the majority group is show in Table 1. The quantiative results in Table 2 indicate an improvement of $9\%$ over the Group DRO (Partial) method for the proposed Worst-off DRO algorithm. The Group DRO (Oracle) method achieves the highest minority group accuracy of $80\%$. The minority group performance for the ERM method, with an accuracy of $45\%$, is comparable to Group DRO (Partial). All the methods are similar in terms of the average group accuracy with values $> 90\%$.

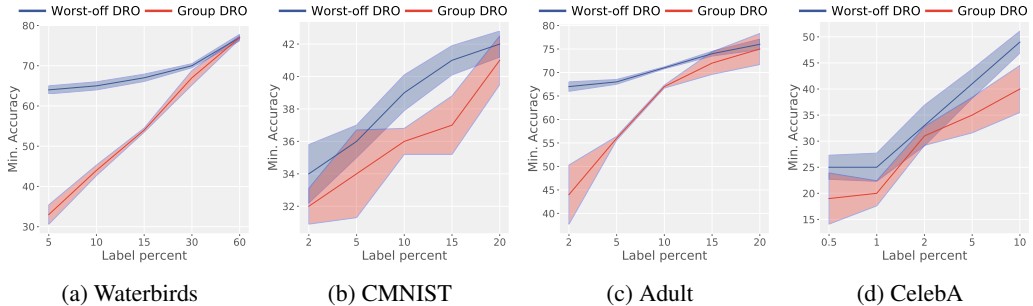

| (a) Waterbirds | (b) CMNIST | (c) Adult | (d) CelebA |

Figure 2: **Increasing the labelled samples.** Minority group accuracies are plotted at different counts of the labelled samples in the training dataset. Both, Group DRO (Partial) and Worst-off DRO algorithms improve the minority group accuracies with more training labels. Also, the Worst-off DRO method has higher accuracy values than Group DRO method. The aggregate group accuracies are shown in the Appendix Figure 5.

### 4.3 ABLATION STUDIES

In this section, we discuss different components of our algorithm that influences it's performance.

#### 4.3.1 INCREASING THE LABELLED SAMPLES.

Recall that for the quantitative results in Table 2, the number of labelled samples were around $10\%$ of the total training samples. In this section, we investigate the effects of increasing the number of labelled samples provided to the training algorithm. Although obtaining annotations for groups is an arduous task [18], having more labelled groups provides two benefits for the algorithm. Firstly, the standard deviation of errors in estimating the marginal probabilities from the labelled portion of the data reduces [36] ( $\approx \sqrt{\# \text{ samples}}$ rate). Secondly, labelled groups reinforce an accurate evaluation of the Rawlsian objective in Equation 2 and appropriate weight updates for the groups. The results shown in Figure 2 depict the minority group accuracy at different labelled percent thresholds. The corresponding plots for average group accuracies are provided in the Appendix Figure 5. Worst-off DRO method is compared with Group DRO (Partial). It is observed that for both the methods, the minority group's accuracy increases with more labelled data. Furthermore, the accuracy values for Worst-off DRO method are better than Group DRO at several thresholds. The methods converge at a threshold specific to the datasets. Increasing the labelled counts beyond such a threshold saturates the Worst-off DRO performance, however, Group DRO (Partial) consistently improves until Oracle performance is attained.

#### 4.3.2 MINORITY GROUP VS. OVERALL ACCURACY.

As discussed in Section 1, several groups in the training dataset, especially the minority groups, could be distributionally different from the majority group samples. Consequently, a mild tradeoff surfaces between the minority group accuracy values and the aggregate accuracies. Addressing this issue, recall that we leverage a robust model selection criterion such as NVP (see Section 4, model selection paragraph) that balances both the minority and aggregate group accuracies. We extend the results in this section, by plotting evaluations at different hyper-parameter choices for our algorithm. Figure 3 contrasts minority group and overall accuracy across all the datasets. Evidently, the top-right corners are desirable regions for the models to be present with maximum performance across both the metrics. The Adult dataset in Figure 3 shows a clear envelope on the Worst-off DRO models that surpass the corresponding Group DRO models. Similar trend exists on the remaining datasets with more Worst-off DRO models concentrated in the top-right corner.

#### 4.3.3 PROGRESSION OF GROUP WEIGHTS

Our algorithm 1, proceeds by assigning weights, the $q-$value's, to every group. These $q-$values are updated through exponential ascent similar to [30]. Noticeably, the updates on $q$ depend on the worstoff group assignments determined from the constraint set $\mathcal{C}_{\bar{\mathbf{P}}, \delta + \epsilon}$. In this section, we investigate on how these group weights evolve through the several iteration of the proposed algorithm. Figure 4 plots this evolution across different datasets with the number of curves in a given plot matching the group count of that dataset. Since the $q-$ weights are initialized uniformly at random, the curves begin at the same value. As the training progresses, it is observed that the weights on the minority

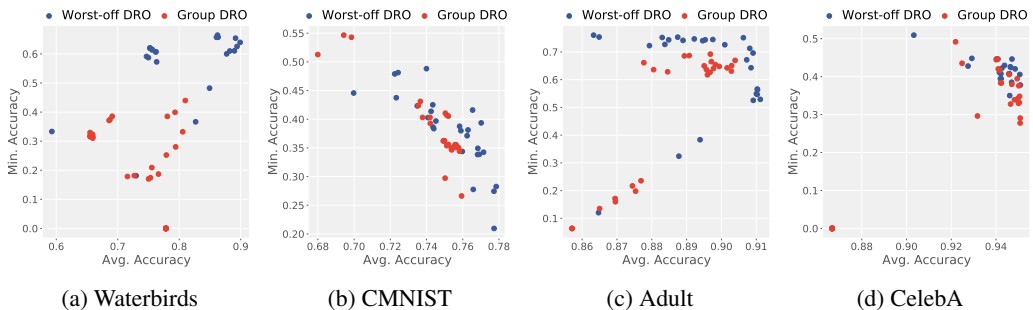

(a) Waterbirds    (b) CMNIST    (c) Adult    (d) CelebA

Figure 3: **Minority Group vs. Average Accuracy.** Evaluations for different hyper-parameter choices are plotted for Worst-off DRO and Group DRO (Partial) methods. Models from Worst-off DRO training are concentrated in the top-right corner of the plots. This is desirable indicating a high accuracies across the two metrics. For model selection from among the possible choices, we adopt the NVP procedure (see Section 4).

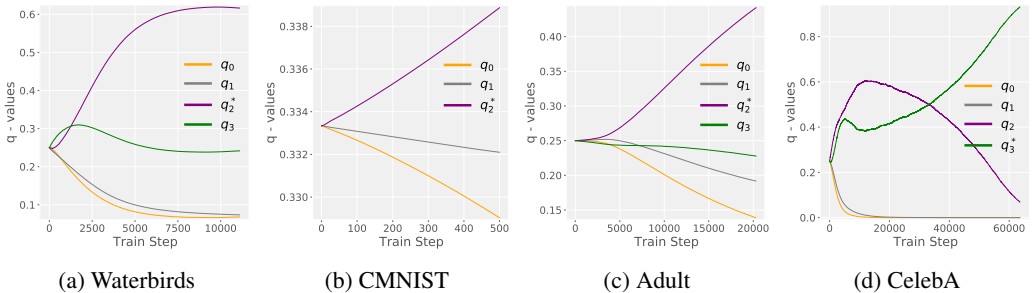

(a) Waterbirds    (b) CMNIST    (c) Adult    (d) CelebA

Figure 4: **Progression of group weights.** The evolution of $q-$values (see Algorithm 1) is plotted for each group. The $q-$values for the minority groups increases gradually while those of the majority groups reduce. A high $q-$value indicates that the corresponding group receives a higher weight relative to other groups. In the plots, minority group is indicate by a $*$ on $q$.

groups gradually increase and those on the majority groups reduce. The plots indicate a high $q-$ value on the minority groups towards the end of training. This is desirable because the empirical risk on the minority groups getting upweighted relative to the majority groups.

## 5 CONCLUSION

We present Worst-off DRO, an invariant learning method across groups when partial group labels are available. The formulation of Worst-off DRO extends that of Group DRO by optimizing the loss against the worst-off group assignments in the constraint set. By reducing the constraint set with the marginal distribution, we reduce the optimization parameter space while keeping the objective to be an upper bound to that of the Group DRO with true group assignments with high probability. By harnessing both labeled and unlabeled data in terms of group, we demonstrate in experiments that the Worst-off DRO outperforms both ERM, UnsupDRO, which do not make use of available group labels, as well as the Group DRO (Partial), which does not use unlabeled data.

One future direction, when marginal distribution is not available, is to relax our *missing completely at random* assumption and bring in different but more realistic modeling assumptions on the missingness of group labels, such as missing at random (MAR), where missing values depend on other observed attributes [12]. In addition, it would be valuable to design a reduced constraint set containing the true group assignment to reduce the performance gap to the Group DRO (Oracle).

## ETHICS STATEMENT

Machine Learning (ML) models that perform poorly on a minority group or environment have raised a lot of concerns within the AI community and broader society in recent years. To democratize ML in real world, learning ML models that perform robustly across groups or environments has become an important venue of research. The proposed Worst-off DRO is a versatile method that could be employed to train an invariant classifier across groups even when the group information is available only for the portion of the data. This is a rather practical scenario as the group information could be missing for various reasons during the data collection. We further emphasize the importance of theoretical result showing the objective of Worst-off DRO being an upper bound to that of Group DRO with complete group information for safety-critical ML applications.

## REPRODUCIBILITY

We write our experimental code from scratch using PyTorch library [26]. Due to its similarity, our implementation may closely follow that of Group DRO [30].[1] One of the key differentiation of Worst-off DRO is the inner maximization solver for the worst-off group assignments $\{\hat{g}\}$, which we elaborate the exact code using CVXPY solver [8] in Algorithm 2 of Appendix. Additional implementation details, including the neural network architectures, as well as value for hyperparameters including the learning rate, weight decay, batch size, number of training epochs, and algorithm-specific parameters are summarized in Table 3 of Appendix and Section A.3 and A.5.

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

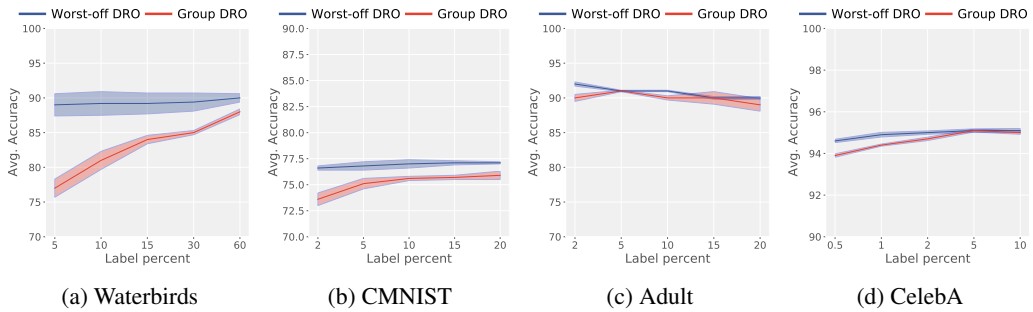

(a) Waterbirds     (b) CMNIST     (c) Adult     (d) CelebA

Figure 5: **Increasing the labelled samples - Average Group Accuracy.** We plot the average group accuracies as a function of labelled samples. These accuracies remain fairly similar as the count of labelled samples grows.

# A APPENDIX

## A.1 PROOF OF LEMMAS

**Lemma 1.** *The constraint set $\mathcal{C}_{\mathbf{p}^\star,\epsilon}$ contains the true group labels $\{g_i^\star\}_{i=1}^N$ with high probability:*

$$P(\{g_i^\star\}_{i=1}^N \in \mathcal{C}_{\mathbf{p}^\star,\epsilon}) \geq 1 - 2e^{-2N\epsilon^2}$$

*Proof.* The probability of the true group assignment $\{g_i^\star\}_{i=1}^N$ in the constraint set $\mathcal{C}_{\mathbf{p}^\star,\epsilon}$ is written as follows:

$$P(\{g_i^\star\}_{i=1}^N \in \mathcal{C}_{\mathbf{p}^\star,\epsilon}) = P\left(\left|p_j^\star - \frac{1}{N}\sum_{i=1}^N \mathbb{1}\{g_i^\star = j\}\right| \leq \epsilon\right) \geq 1 - 2e^{-2N\epsilon^2} \tag{10}$$

where Equation 10 holds true from Hoeffding's inequality. $\square$

**Lemma 2.** *The constraint set $\mathcal{C}_{\bar{\mathbf{p}},\delta+\epsilon}$ contains the true group labels $\{g_i^\star\}_{i=1}^N$ with high probability:*

$$P(\{g_i^\star\}_{i=1}^N \in \mathcal{C}_{\bar{\mathbf{p}},\delta+\epsilon}) \geq 1 - 2e^{-2N\epsilon^2} - 2e^{-2K\delta^2}$$

*Proof.* Using Hoeffding's inequality, we can show that the estimation error of the marginal distribution is bounded by $\delta$ with high probability as follows:

$$P(|p_j^\star - \bar{p}_j| \leq \delta) = P\left(\left|p_j^\star - \frac{1}{K}\sum_{i=1}^K \mathbb{1}\{g_i^\star = j\}\right| \leq \delta\right) \geq 1 - 2e^{-2K\delta^2} \tag{11}$$

Furthermore, we show using Hoeffding's inequality that

$$P\left(\left|p_j^\star - \frac{1}{N}\sum_{i=1}^N \mathbb{1}\{g_i = j\}\right| \leq \epsilon\right) \geq 1 - 2e^{-2N\epsilon^2} \tag{12}$$

Now, the probability of the true group assignment $\{g_i^\star\}_{i=1}^N$ in the constraint set $\mathcal{C}_{\bar{\mathbf{p}},\delta+\epsilon}$ is written as follows:

$$P(\{g_i^\star\}_{i=1}^N \in \mathcal{C}_{\bar{\mathbf{p}},\delta+\epsilon}) = P\left(\left|\bar{p}_j - \frac{1}{N}\sum_{i=1}^N \mathbb{1}\{g_i = j\}\right| \leq \delta + \epsilon\right) \tag{13}$$

$$\geq P\left(\left\{|p_j^\star - \bar{p}_j| \leq \delta\right\} \cap \left\{\left|p_j^\star - \frac{1}{N}\sum_{i=1}^N \mathbb{1}\{g_i = j\}\right| \leq \epsilon\right\}\right) \tag{14}$$

$$\geq P\left(|p_j^\star - \bar{p}_j| \leq \delta\right) + P\left(\left|p_j^\star - \frac{1}{N}\sum_{i=1}^N \mathbb{1}\{g_i = j\}\right| \leq \epsilon\right) - 1 \tag{15}$$

$$\geq 1 - 2e^{-2K\delta^2} - 2e^{-2N\epsilon^2} \tag{16}$$

where Equation 14 is due to that the intersection of events in Equation 14 is a subset of an event in Equation 13, and Equation 15 is derived using union bound. $\square$

| | Waterbirds | | CMNIST |
|---|---|---|---|
| Learning Rate | $0.0001, 0.00001, 0.000001$ | Learning Rate | $0.001, 0.0001, 0.00001$ |
| Weight Decay | $1.5, 1.0, 0.1$ | Weight Decay | $0.01, 0.001, 0.0001$ |
| $\eta_{\text{UDRO}}$ | $0.9, 0.8, 0.7, 0.6, 0.5, 0.4, 0.3$ | $\eta_{\text{UDRO}}$ | $0.9, 0.8, 0.7, 0.6, 0.5, 0.4, 0.3$ |
| $\eta_{\text{GDRO}}$ | $0.1, 0.01, 0.001$ | $\eta_{\text{GDRO}}$ | $0.01, 0.001, 0.0001$ |
| $\eta_{\text{WDRO}}$ | $0.1, 0.01, 0.001$ | $\eta_{\text{WDRO}}$ | $0.01, 0.001, 0.0001$ |

| | Adult | | CelebA |
|---|---|---|---|
| Learning Rate | $0.001, 0.0001, 0.00001$ | Learning Rate | $0.0001, 0.00001, 0.000001$ |
| Weight Decay | $0.01, 0.001, 0.0001$ | Weight Decay | $1.0, 0.1, 0.01$ |
| $\eta_{\text{UDRO}}$ | $0.9, 0.8, 0.7, 0.6, 0.5, 0.4, 0.3$ | $\eta_{\text{UDRO}}$ | $0.9, 0.8, 0.7, 0.6, 0.5, 0.4, 0.3$ |
| $\eta_{\text{GDRO}}$ | $0.01, 0.001, 0.0001$ | $\eta_{\text{GDRO}}$ | $0.1, 0.01, 0.001$ |
| $\eta_{\text{WDRO}}$ | $0.01, 0.001, 0.0001$ | $\eta_{\text{WDRO}}$ | $0.1, 0.01, 0.001$ |

Table 3: **Grid search for Table 2.** The range of values for each hyper-parameter is listed. A grid search over these hyper-parameters is conducted to identify the best performing model. Models outside these range values were observed to be either unstable or not converging. Model selection is done based on NVP (novel validation procedure) where first the models, with higher overall accuracies, are selected. From the top five such performing models, the one with the highest minority group accuracy is picked.

| Dataset | Method | Architecture | Learning Rate | Weight Decay | Batch Size | # Epochs | Other params |
|---|---|---|---|---|---|---|---|
| Waterbirds | ERM | ResNet50 | 0.0001 | 0.1 | 128 | 300 | - |
| Waterbirds | Unsup DRO | ResNet50 | 0.0001 | 0.1 | 128 | 300 | $\eta=0.3$ |
| Waterbirds | Group DRO-(Oracle) | ResNet50 | 0.00001 | 1.0 | 128 | 300 | $\eta=0.001$ |
| Waterbirds | Group DRO-(Partial) | ResNet50 | 0.00001 | 0.1 | 128 | 300 | $\eta=0.001$ |
| Waterbirds | Worst-off DRO | ResNet50 | 0.00001 | 1.0 | 128 | 300 | $\eta=0.001$ |
| CMNIST | ERM | MLP(390,390) | 0.001 | 0.01 | - | 500 | - |
| CMNIST | Unsup DRO | MLP(390,390) | 0.00001 | 0.001 | - | 500 | $\eta=0.4$ |
| CMNIST | Group DRO-(Oracle) | MLP(390,390) | 0.0001 | 0.001 | - | 500 | $\eta=0.001$ |
| CMNIST | Group DRO-(Partial) | MLP(390,390) | 0.001 | 0.01 | - | 500 | $\eta=0.001$ |
| CMNIST | Worst-off DRO | MLP(390,390) | 0.0001 | 0.01 | - | 500 | $\eta=0.0001$ |
| Adult | ERM | MLP(64,32) | 0.0001 | 0.001 | 128 | 200 | - |
| Adult | Unsup DRO | MLP(64,32) | 0.0001 | 0.001 | 128 | 200 | $\eta=0.3$ |
| Adult | Group DRO-(Oracle) | MLP(64,32) | 0.0001 | 0.001 | 128 | 200 | $\eta=0.0001$ |
| Adult | Group DRO-(Partial) | MLP(64,32) | 0.0001 | 0.01 | 128 | 200 | $\eta=0.001$ |
| Adult | Worst-off DRO | MLP(64,32) | 0.00001 | 0.001 | 128 | 200 | $\eta=0.0001$ |
| CelebA | ERM | ResNet50 | 0.0001 | 0.01 | 128 | 50 | - |
| CelebA | Unsup DRO | ResNet50 | 0.0001 | 0.01 | 128 | 50 | $\eta=0.6$ |
| CelebA | Group DRO-(Oracle) | ResNet50 | 0.00001 | 0.1 | 128 | 50 | $\eta=0.1$ |
| CelebA | Group DRO-(Partial) | ResNet50 | 0.00001 | 0.01 | 128 | 50 | $\eta=0.1$ |
| CelebA | Worst-off DRO | ResNet50) | 0.00001 | 0.1 | 128 | 50 | $\eta=0.001$ |

Table 4: **Hyperparamter choices for Table 2.** We list the hyper-parameters selected using the NVP procedure (see Section 4) after performing grid-search. Learning rate and weight decay are an important set of parameters that influences the minority group performance. Each baseline has it's algorithm-specific hyper-parameter such as step-size of the simplex weights in Group DRO ($\eta_{\text{GDRO}}$), the loss threshold in Unsup DRO ($\eta_{\text{UDRO}}$) and the step size for the group weights in Worst-off DRO ($\eta_{\text{WDRO}}$). The symbol "-" for batchsize in CMNIST experiments indicate the use of full-batch data for training.

## A.2   NOTES ON OPTIMIZATION

When using CVXPY to solve for the Worst-off DRO assignments, we simplify the problem by replacing the data marginal distribution $\sum_{i=1}^{N} \hat{g}_{ij}$ in the denominator of Equation 9 to $\bar{\mathbf{p}}_j$, thus providing us with a convex optimization problem. The code for the solver is available in Algorithm 2.

Next, Using three sample, we provide an example of the worst-off assignments made by our algorithm,

**Example 3.** *Consider three samples with loss values $l_1 > l_2 > l_3$ and two predefined groups. Assume the marginal probabilities $\bar{\mathbf{p}}_1 = 0.6$ and $\bar{\mathbf{p}}_2 = 0.4$. Without loss in generality, assume $\frac{q_1}{\bar{\mathbf{p}}_1} > \frac{q_2}{\bar{\mathbf{p}}_2}$. Solving for Worst-off DRO objective results in the following group assignments. With*

---

**Algorithm 2** Group Assignment Solver using CVXPY library

---

```python
import cvxpy as cp
import numpy as np

class Solver(object):
  def __init__(self, n_controls, bsize, marginals, epsilon, labeled=None):
    """Group assignment solver.

    Arguments:
      n_controls: An integer for the number of groups.
      bsize: An integer for the batch size.
      marginals: A 2D array for the marginal distribution.
      epsilon: A float for the variance.
      labeled: A tuple for labeled data indices and their value.
    """
    self.X = cp.Variable((bsize, n_controls))
    self.l = cp.Parameter((bsize, 1))
    self.p = cp.Parameter((n_controls, 1), value=marginals)
    self.q = cp.Parameter(n_controls)
    if labeled is not None:
      labeled_idx, labeled_value = labeled
    counts = cp.sum(self.X, axis=0, keepdims=True)

    obj = ((self.l.T @ self.X) / self.p.T) @ self.q
    constraints = [self.X >= 0,
                   cp.sum(self.X, axis=1, keepdims=True) == np.ones((bsize, 1)),
                   cp.abs(cp.sum(self.X, axis=0, keepdims=True) / bsize - self.p.T) <=
                       epsilon]
    if labeled is not None:
      constraints += [self.X[labeled_idx] == labeled_value]

    self.prob = cp.Problem(cp.Maximize(obj), constraints)

  def cvxsolve(self, losses, weights):
    """Solver.

    Arguments:
      losses: A 2D array for loss values.
      weights: A 1D array for group weights q.

    Returns:
      A 2D array for soft group assignments.
    """
    self.l.value = losses
    self.q.value = weights
    self.prob.solve()
    return self.X.value
```

---

*constraint* $\mathcal{C}_{\bar{\mathbf{p}},\epsilon=0}$, *we obtain* $\{\hat{g}^t\} = \begin{pmatrix} 1 & 0 \\ 0.8 & 0.2 \\ 0 & 1 \end{pmatrix}$. *Here, the $i^{th}$ row indicates the assignment given to sample $l_i$.*

The above example can be derived by identifying a $\{\hat{g}\}$ that satisfies the constraints $\sum_{i=1}^{N} \hat{g}_{i1} \leq N\bar{\mathbf{p}}_1$ and $\sum_{i=1}^{N} \hat{g}_{i2} \leq N\bar{\mathbf{p}}_2$, where $N = 3, \bar{\mathbf{p}}_1 = 0.6$ and $\bar{\mathbf{p}}_1 = 0.4$ and correspondingly maximizes Worst-off DRO objective. The above example informs us that group assignments depend on the magnitude of loss values in addition to the group weights and marginal probabilities. Importantly, we find that *high loss samples are assigned to groups with high group weights and low marginal probabilities*.

Observe that the marginal constraints form a key ingredient of our algorithm as per the above example. Without the marginal constraints, we would have obtained the group assignments $\{\hat{g}^t\} = \begin{pmatrix} 1 & 0 \\ 1 & 0 \\ 1 & 0 \end{pmatrix}$. That is, the assignments would have been made independent of the loss values and sparsely restricted to the group with large $\frac{q_j}{\bar{\mathbf{p}}_j}$ value.

### A.3 HYPER-PARAMETER TUNING

Hyper-parameters were selected for each algorithm by performing an NVP procedure (see Section 4). The best performing model was identified on the validation set associated with each dataset.

| | Waterbirds | | CMNIST | | Adult | | CelebA | |
|---|---|---|---|---|---|---|---|---|
| | **min** | **avg** | **min** | **avg** | **min** | **avg** | **min** | **avg** |
| Group DRO (Oracle) | 0.45 | 0.02 | 0.57 | 0.33 | 0.94 | 0.87 | 1.31 | 0.27 |
| ERM | 0.99 | 0.07 | 1.47 | 0.49 | 1.06 | 0.28 | 3.02 | 0.05 |
| Unsup DRO | 0.89 | 0.12 | 1.39 | 0.64 | 1.73 | 0.18 | 1.81 | 0.02 |
| Group DRO (Partial) | 1.25 | 1.25 | 0.76 | 0.20 | 0.32 | 0.34 | 4.54 | 0.08 |
| Worst-off DRO | 0.96 | 0.17 | 1.14 | 0.35 | 0.21 | 0.13 | 2.05 | 0.10 |

Table 5: **Quantitative Results - Standard Deviations.** The standard deviations over three random runs of Table 2 is provided. For baselines, we consider an ERM, Unsup DRO [14], Group DRO (Partial) for partly labelled Group DRO [30] method, Group DRO (Oracle) for the fully supervised model.

All the measures were computed and averaged over three random runs. A list of all the hyper-parameters that were tuned for are available in Table 3. The final hyper-parameters selected for each method can be viewed from Table 4.

### A.4 Additional Experimental Results

We provide the following additional results, first, in Figure 5, we show average group accuracies as a function of labelled sample counts. The average group accuracies of the Worst-off DRO method are closely similar across various labelled sample counts. The Group DRO method shows a slight increasing trend in the average accuracies as the number of labelled samples increase. Next, corresponding to the quantitative results of Table 2 in the paper, we provide standard deviations of those results in Table 5. The standard deviations for all the methods are comparable.

### A.5 More details on the datasets

#### A.5.1 Waterbirds

This dataset was first introduced in [30] and has been developed by cropping images of birds from the CUB dataset [34] and pasting them on the backgrounds from the Places dataset [41]. A ResNet50 model, pre-trained with ImageNet weights, has been used for training in experiments on this dataset. No data augmentation has been applied for any of the Algorithms.

#### A.5.2 CMNIST

CMNIST dataset comprised of two groups of MNIST images each with a specific color. As per the description in the main paper, the target label is flipped with a specific correlation to the color. Following the implementation of [7], the digit images contain two channels and were downsampled to $14 \times 14$ pixels.

#### A.5.3 Adult

The Adult dataset used in the paper was obtained from the UCI repository [10]. It contains $44,842$ samples. The features that were used in the experiments include "age", "workclass", "fnlwgt", "education", "education-num", "marital-status", "occupation", "relationship", "race", "sex", "capital-gain", "capital-loss", "hours-per-week", "native-country", "income". A positve target label in this dataset is indicated by the attribute "income-bracket" being above $50K\$$.

#### A.5.4 CelebA

For this dataset, the official train-val-test splits as recommended by [21] has been used. Similar to the Waterbirds experiments, a pre-trained ImageNet-based ResNet50 model has been used for the implementations.

### A.6 Ablation study on increasing the constraint set size.

We conduct experiments on Worst-off DRO method for different values of the $\epsilon$ parameter in the set $\{0, 0.001, 0.01, 0.1, 1\}$. The test set accuracies on the minority group and average group are

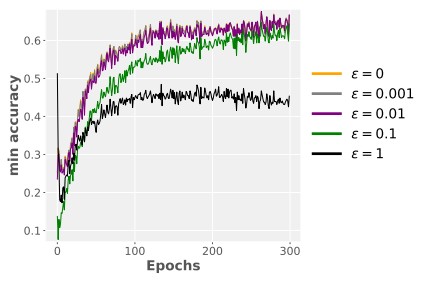

(a) Waterbirds Minority Group Accuracy

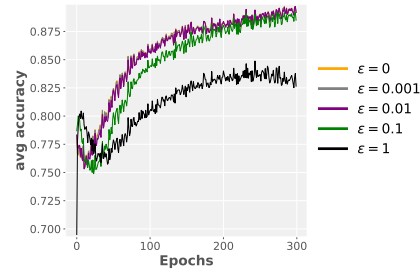

(b) Waterbirds Average Group Accuracy

Figure 6: **Varying the $\epsilon$ parameter in the constraints.** The marginal constraint is gradually relaxed by increasing the $\epsilon$ parameter. The accuracies in the plots are computed on the test sets. Performance of the models with $\epsilon \leq 0.01$ are similar, however, the accuracies drop when increasing $\epsilon$ beyond $0.01$ threshold.

reported in Figure 6. Increasing the $\epsilon$ value also increases the constraint set size because the marginal constraint is gradually relaxed. Figure 6 shows that the both minority group accuracy and average group accuracy values reduce with increase in $\epsilon$ value beyond $0.1$ threshold. The accuracy values for $\epsilon \leq 0.01$ are comparable. A similar trend hold on other datasets as well.

