# OpenReview forum: "Invariant Learning with Partial Group Labels"
_ICLR.cc/2022/Conference — ICLR 2022 Submitted_

### Official Review · Reviewer_RXdi · 2021-10-31

**Correctness:** 4
**Technical Novelty And Significance:** 2
**Empirical Novelty And Significance:** 2
**Recommendation:** 6
**Confidence:** 3

**Main Review:**

PROS:
- This paper focus on an interesting problem that has both practical and theoretical importance. In general, learning a fairness-aware classifier with weak supervision is an important problem.
- The extension to Group DRO is intuitive and reasonable. As stated in the paper, it naturally forms an upper bound of Group DRO.
- The paper is overall well-written and easy to follow. I can get the point authors want to convey without ambiguity.

CONS:
- How the size of constraint set affects the convergence / efficiency is not stated theoretically / empirically. It is a crucial property for the proposed algorithm to be feasible.
- Estimation of the true marginal distribution plays an important role in constraining the group assignment set, and this should be emphasized in the paper. Corresponding empirical verification for Lemma 2 is missing. Also, how the algorithm will perform if the estimation is wrong, thus the true p is not in the set?


**Summary Of The Paper:**

This paper tackles the problem of optimization for the Rawlsian criterion with only partial group labels. They propose to optimize according to the worst case of group assignment and also propose a method to constrain the set of group assignment.


**Summary Of The Review:**

Due to the weakness written in the main review, I choose to give a weak reject.

---

> ### Author Response · Authors · 2021-11-17
> **Response to Reviewer RXdi**
>
> We are happy that the reviewer finds our paper to solve an interesting problem with practical and theoretical importance.
>
> ---
>
>
> **Question** How the size of constraint set affects the convergence / efficiency is not stated theoretically / empirically. It is a crucial property for the proposed algorithm to be feasible.
>
> **Answer** We have added an ablation study in the Appendix $A.6$ of the paper that varies the \epsilon parameter of the algorithm. Increasing $\epsilon$ increases the size of the constraint set. As we observe from the plots, the worstoff group accuracy gradually reduces when $\epsilon$ is increased with the best value of $\epsilon$ being $0.001$.
>
> ---
>
> **Question** Estimation of the true marginal distribution plays an important role in constraining the group assignment set, and this should be emphasized in the paper. Corresponding empirical verification for Lemma 2 is missing.
>
> **Answer** In our experiments p* is unknown apriori. $\bar p$ is computed from the partially labelled dataset. Hence, all our experiments constitute the setting of Lemma 2.
>
> ---
>
> **Question** Also, how the algorithm will perform if the estimation is wrong, thus the true p is not in the set?
>
> **Answer**  We restate the response given to “Reviewer sood” with regards to this question.
>
> When $\bar p$ is misspecified, our algorithm naturally provides a mitigation strategy. To start with, the littles test ([ref]) can be used to determine if the data is not MCAR in which case it’s likely for $\bar p$ to be misspecified. In this setting,increasing the $\epsilon$ hyper-parmeter provides more flexibility with the choice of assignments (please see an updated Appendix section for an ablation study on \epsilon values). On the theoretical side, our Lemma 2 captures the error in misspecification of $\bar p$ in relation to $p*$ via the $\delta$ gap (Equation 11 in Appendix.)
>
> [ref] Little, Roderick JA. "A test of missing completely at random for multivariate data with missing values." Journal of the American Statistical Association 83.404 (1988): 1198-1202.

---

> > ### Comment · Reviewer_RXdi · 2021-11-25
> > **Thanks for author response**
> >
> > I appreciate that the authors respond to reviewers comments in detail. My concerns over constraint set size and estimation of p are partially  addressed. I also went through other other comments and corresponding response. I would like to raise my score but will not champion it.

---

> > > ### Author Response · Authors · 2021-11-28
> > > **Thanks!**
> > >
> > > We thank the reviewer for going through our answers. We appreciate the increase in the score!

---

### Official Review · Reviewer_aqwr · 2021-11-01

**Correctness:** 4
**Technical Novelty And Significance:** 3
**Empirical Novelty And Significance:** 3
**Recommendation:** 6
**Confidence:** 3

**Main Review:**

##########################################

Pros:

+ The theories in this paper are solid and are verified to be right.

+ This paper addresses a more realistic setting about fairness, and their solution seems to make sense.

##########################################

Cons:

- The research question of this paper is clear. However, after intruducing this question, they directly turn to explain Worst-off DRO in detail. This is too abrupt. They should add at least one paragraph about its motivation.

-  They convert the feasible set to $\mathcal{C}_{\bar{p},\epsilon}$ by adding an extra condition, thus giving an error estimation will be better.

- Pseudo label-based methods serve as important solutions for this research question. However, related baselines are missing.

- If possible, more fairness-related datasets, e.g., Arrest, Violent, German, should be included in experiments. (Not necessary in the rebuttal period)

##########################################

Typo:

1. "We optimize for the a worstoff soft group assignment" -- "the" or "a" ?

and so on...

##########################################

Questions during rebuttal period:

Please address and clarify the cons above.

#########################################



**Summary Of The Paper:**

This paper aims to learn invariant representations for group membership using partially group labeled data. Thus, they propose the Worst-off DRO, which extends Group DRO by optimizing the loss against the worst-off group assignments in the constraint set.

**Summary Of The Review:**

This paper works on a more practial fairness scenario and gives a solid solution in principle. However, there still exists some issues about writting and minor technical details. If these issues can be solved well in rebuttal phase, I am willing to support this paper.

---

> ### Author Response · Authors · 2021-11-17
> **Response to Reviewer aqwr**
>
> We are thankful that the reviewer considers our work to be a solid solution and addresses realistic fairness settings.
>
> ---
>
> **Question** Pseudo label-based methods serve as important solutions for this research question. However, related baselines are missing.
>
> **Answer** Since we discuss and propose our method for fairness applications, we feel that pseudo-labeling methods could raise important ethical concerns. For example, the groups in Adult dataset are sex and gender. Inferring these group labels with pseudo-label based methods is harmful as the estimated labels could be likely mis-used . We elaborate on this point in Reviewer Jwqz’s response. We will proceed with the inclusion of pseudo-label baselines as per reviewers recommendation.
>
> ---
>
> **Question** If possible, more fairness-related datasets, e.g., Arrest, Violent, German, should be included in experiments. (Not necessary in the rebuttal period)
>
> **Answer** Thanks for the suggestion on these datasets!
>
> ---
>
> **Question** They convert the feasible set to $C_{p, \epsilon}$ by adding an extra condition, thus giving an error estimation will be better.
>
> **Answer** We request the reviewer to elaborate this point. From Section 3.2 of the paper, we find that $C_{p, \epsilon} \subset C$ due to the marginal constraints. When $p=p^*$, we observe as a consequence of Lemma1 that the GroupDRO loss is more tightly upper bounded by Worstoff-DRO loss (see Equation $6$ in the paper).

---

> > ### Comment · Reviewer_aqwr · 2021-11-25
> > **Thanks for your response**
> >
> > I have carefully read the response and the other reviewers' comments, and my concerns have been addressed well. I hope the authors merge related discussion about $\hat{p}$ and include faireness datasets into their paper. I am willing to increase my score to 6.

---

> > > ### Author Response · Authors · 2021-11-28
> > > **Thanks!**
> > >
> > > We are glad that the reviewer's comments were addressed in our response and appreciate the increase in score. We will incorporate their comments in our manuscript.

---

### Official Review · Reviewer_sood · 2021-11-02

**Correctness:** 2
**Technical Novelty And Significance:** 2
**Empirical Novelty And Significance:** 1
**Recommendation:** 3
**Confidence:** 4

**Details Of Ethics Concerns:**

The paper proposes to address bias and improve worst-group performance. However, I am concerned that the unrealistic *missing completely at random (MCAR)* assumption casts doubts on the validity of the proposed method, without the authors providing significant justification or mitigation strategies.

**Main Review:**

The proposed method is technically sound, nicely polished, and well presented.

However, I have two major concerns:

1. The paper is motivated by real-world scenarios where only partial group information is available, for example when individuals may choose not to reveal this information due to privacy concerns. These realistic situations should usually result in a *biased* distribution of which samples contain the group annotation, typically due to demographics leading to unknown individual cohorts. However, the entire paper is based on the assumption that group information is *missing completely at random (MCAR)*, in both theoretical section and experimental section. I strongly advise that the authors carefully study other types of missingness, and build ways to understanding the impact of group information availability bias on the learning algorithm proposed. For instance, the algorithm depends on estimating the group distribution $\hat{\mathbf{p}}$ through the available samples, this estimation can be biased if not MCAR, which may render the Lemma 2 unreliable at least. Empirical evidence is also called for when $\hat{\mathbf{p}}$ is misspecified.
2. The paper is missing *very important baselines* from SOTA unsupervised methods for improving worst-group performance, e.g., EIIL/JTT. Ideally, the proposed Worst-Off DRO should perform between EIIL/JTT (lower anchor) and Group DRO (anchor bound), since it requires *more* information than any unsupervised method (admittedly, JTT actually does require a small annotated validation set). But since the paper did not report these comparisons, I can only judge from the numbers across those papers on the same dataset (e.g., CUB), and it seems that Worst-Off DRO performs quite poorly compared to JTT for instance. If this is the case, the authors should list those baselines, and discuss the pros/cons as to the advantages of the proposed methods compare to EIIL/JTT.

**Summary Of The Paper:**

The paper proposes a new variant of DRO, called Worst-Off DRO, to address a ubiquitous real-world setting where group labels are only partially available over the training set. The core idea is to introduce a nested optimization to maximize the worst-off group assignment over the entire training set, given some sensible constraints, hence enabling to alternate the standard DRO optimization. Empirical results are reported to support the proposed methods.

**Summary Of The Review:**

The paper is technically sound and well written. However, due to the two major concerns above (unrealistic MCAR assumption and missing important baselines), I can hardly recommend the paper as ready for publication.

---

> ### Author Response · Authors · 2021-11-17
> **Response to Reviewer sood**
>
> We thank the reviewer for their comments and glad that they find our work technically sound, nicely polished, and well presented.
>
> ---
>
> **Question** The paper is motivated by real-world scenarios where only partial group information is available, for example when individuals may choose not to reveal this information due to privacy concerns. These realistic situations should usually result in a biased distribution of which samples contain the group annotation, typically due to demographics leading to unknown individual cohorts. However, the entire paper is based on the assumption that group information is missing completely at random (MCAR), in both theoretical section and experimental section. I strongly advise that the authors carefully study other types of missingness, and build ways to understanding the impact of group information availability bias on the learning algorithm proposed. For instance, the algorithm depends on estimating the group distribution
>  through the available samples, this estimation can be biased if not MCAR, which may render the Lemma 2 unreliable at least. Empirical evidence is also called for when $\bar p$
>  is misspecified.
>
> **Answer** When the data is Missing at Random (MAR), we could estimate the propensity of missingness from other features; then use inverse propensity weighting to get a consistent estimate of the fraction of samples in each group (see [ref1] for example). Alternatively, if provided with the knowledge of the data-generation process, the core effort in extending our method simply involves using off-the-shelf estimators to characterize the probability distributions (see [ref2] for example).
>
> When $\bar p$ is misspecified, our algorithm naturally provides a mitigation strategy. To start with, the littles test ([ref3]) can be used to determine if the data is not MCAR in which case it’s likely for $\bar p$ to be misspecified. In this setting, increasing the $\epsilon$ hyper-parameter provides more flexibility with the choice of assignments (please see an updated Appendix section $A.6$ for an ablation study on $\epsilon$ values). On the theoretical side, our Lemma 2 captures the error in misspecification of $\bar p$ in relation to $p^*$ via the $\delta$ gap (Equation 11 in Appendix.)
>
> [ref1] Zhao, Yuxuan, and Madeleine Udell. "Matrix completion with quantified uncertainty through low rank gaussian copula." arXiv preprint arXiv:2006.10829 (2020).
>
> [ref2] Mohan, Karthika, and Judea Pearl. "Graphical models for recovering probabilistic and causal queries from missing data." Advances in Neural Information Processing Systems 27 (2014): 1520-1528.
>
> [ref3] Little, Roderick JA. "A test of missing completely at random for multivariate data with missing values." Journal of the American Statistical Association 83.404 (1988): 1198-1202.
>
> ---
>
> **Question** The paper is missing very important baselines from SOTA unsupervised methods for improving worst-group performance, e.g., EIIL/JTT.
>
> **Answer**  We restate our response given to "Reviewer Jwqz" with regards to the question. We discuss pros/cons of EIIL/JTT and how these baselines could be harmful for fairness based experiments.
>
> The methods mentioned in the review, such as JTT, GEORGE and EIIL, directly estimate the group label for each sample. Since they make point estimates, the probability of pseudo group labels to be the same as the ground-truth should be very low. On the other hand, the proposed Worstoff-DRO defines a constraint set including the ground-truth group labels with a decent chance and optimizes over all possible configurations, so it is more theoretically safe and conservative. Furthermore, directly estimating the group label could be *harmful* in the context of fairness problems, as estimated labels could be misused by a wrongdoer. Recall the motivating example from the introduction where in demographic surveys individuals choose to conceal sensitive information, such as age or gender, due to privacy concerns. Directly predicting such sensitive group information using the above methods is undesirable. We hoped to convey this in our paper when we claimed our method to benefit safety-critical ML modeling, but we will elaborate the related works section to further discuss this point.
>
> In contrast to the above methods, Worstoff-DRO assignments are unrelated to the actual group labels. Despite being unrelated to the group labels, optimizing the DRO loss over these assignments also optimizes the GroupDRO objective as pointed out in the theoretical guarantees. We attribute this point as a strength in our paper.
>
> On the other hand, observe that all the methods JTT, GEORGE and EIIL are two-stage approaches in contrast to our proposed one-stage optimization. Thus, we cut the compute time by *half!*. Moreover, as discussed in Section 3.2 our method enjoys a direct guarantee and relation to the GroupDRO objective unlike the pseudo-labelling methods.

---

> > ### Comment · Reviewer_sood · 2021-11-24
> > **I'm keeping my scores**
> >
> > Many thanks to the authors for carefully addressing my concerns!
> >
> > Regarding the MCAR assumption, I do agree that some of the concerns can be mitigated/addressed in theory or practice by the suggestions by the authors. However, I think they are not sufficiently or clearly discussed in the manuscript in its current state. I suggest that the authors include a dedicated section to discuss these issues. Besides Appendix A.6 on the $\epsilon$ propagated error in one dataset, it would definitely make the paper a lot stronger if the authors could perform extensive experiments to validate the claims about using estimated/misspecified $\hat{p}$ as made in rebuttal.
> >
> > Regarding the empirical performance with SOTA, since the current paper still chooses to use minority group and overall accuracy as evaluation metrics, I still think it's fair to report the performance of JTT/GEORGE/EIIL. It is definitely an interesting point that the authors are making about the ability of Worst-off DRO being a *better* model for safe-critical models in estimating group assignment given partial demographics, it could help devise an appropriate metric to assess these and compare with other SOTA who may not perform well under such metrics. And regarding computation time, it may be a tricky comparison; for instance, JTT does need to do two trainings, but it is implementable with so little overhead with many off-the-shelf modern parallelizable computational tools. As far as the current algorithm goes, it is not quite clear how to adapt Worst-off DRO in a comparable setting.
> >
> > Overall, I'm keeping my scores.

---

> > > ### Author Response · Authors · 2021-11-30
> > > **Thanks for comments.**
> > >
> > > We are thankful to the reviewer for providing suggestions to include a discussion section on estimated/misspecified $\hat p$. Further, it is a nice suggestion to devise a metric specific to safety-critical models.
> > >
> > > With regards to two-stage models like JTT/GEORGE/EIIL, we provide a few additional points of comparison. Firstly, relative to a one-stage model, a two-stage model introduces a new set of hyper-parameters that need to be tuned. In the case of JTT, the first stage epochs $T$ are an example of such additional hyper-parameters. Secondly, in a two-stage model, errors are propagated to the later stages or a failed first stage leads to an unsuccessful second stage. As stated in the discussion with Reviewer Jwqz, the first stage model could fail due to the model overfitting to the training data in the JTT method, similarly in EIIL/GEORGE inaccurate group inference may block second-stage invariant learning besides raising ethical issues on pseudo-label misuse.

---

### Official Review · Reviewer_Jwqz · 2021-11-04

**Correctness:** 4
**Technical Novelty And Significance:** 4
**Empirical Novelty And Significance:** 3
**Recommendation:** 3
**Confidence:** 5

**Main Review:**

Weaknesses:
- The main criticism of the paper is that the empirical results are not strong. The simple baselines of ERM and Group DRO (partial) Specifically, on Waterbirds and CelebA, many prior methods (such as JTT, GEORGE) that do not require *any* group labels substantially outperform this paper in terms of worst-group performance. Similarly, on CMNIST, EIIL does not require group labels and outperforms the proposed approach (and I suspect JTT and GEORGE would as well if evaluated).
- While the theoretical results are correct, they are of limited impact. The gap between L_{GDRO} and L_{WDRO}, even the marginal distribution version, is likely to be large because the constraint set quite is large. In particular, for the points without group labels, only the marginal distributions have to approximately match, but given this the assignment of the individual group labels to the individual points is arbitrary, whereas in reality we could use the data features to better predict the probability of a given point having a particular group label. Thus, L_{WDRO} is a fairly pessimistic loss upper bound whose loss landscape may be quite different than that of L_{GDRO}; indeed, this is borne out by the empirical results.

Suggestions:
- While the current results are not strong, perhaps there are settings in which the relatively pessimistic L_{WDRO} loss is indeed preferable to existing baselines such as those mentioned above. Finding such a dataset would be helpful.
- It would also be very helpful to construct a synthetic task on which L_{WDRO} can be proven to be preferable to baselines such as ERM and Group DRO (Partial).

Strengths:
- The ablation studies are well-conducted and provide useful insight.

**Summary Of The Paper:**

This paper studies the case of group robustness when only some of the group labels are known. They propose a generalized objective for this setting that considers the worst-case possible group assignment subject to the known group labels, and an alternating optimization algorithm for minimizing this objective.

**Summary Of The Review:**

Overall, the paper has some interesting ideas but unfortunately has limitations, and fairly poor performance in practice. My actual score would be 4, but the rating is unavailable.

---

> ### Author Response · Authors · 2021-11-17
> **Response to Reviewer Jwqz**
>
> We thank the reviewer for the detailed suggestions. We are glad that the reviewer finds our paper to contain interesting ideas and well-conducted ablation studies.
>
> ---
>
> **Question** The main criticism of the paper is that the empirical results are not strong. The simple baselines of ERM and Group DRO (partial) Specifically, on Waterbirds and CelebA, many prior methods (such as JTT, GEORGE) that do not require any group labels substantially outperform this paper in terms of worst-group performance. Similarly, on CMNIST, EIIL does not require group labels and outperforms the proposed approach (and I suspect JTT and GEORGE would as well if evaluated).
>
> **Answer** The methods mentioned in the review, such as JTT, GEORGE and EIIL, directly estimate the group label for each sample. Since they make point estimates, the probability of pseudo group labels to be the same as the ground-truth should be very low. On the other hand, the proposed Worstoff-DRO defines a constraint set including the ground-truth group labels with a decent chance and optimizes over all possible configurations, so it is more theoretically safe and conservative. Furthermore, directly estimating the group label could be *harmful* in the context of fairness problems, as estimated labels could be misused by a wrongdoer. Recall the motivating example from the introduction where in demographic surveys individuals choose to conceal sensitive information, such as age or gender, due to privacy concerns. Directly predicting such sensitive group information using the above methods is undesirable. We hoped to convey this in our paper when we claimed our method to benefit safety-critical ML modeling, but we will elaborate the related works section to further discuss this point.
>
> In contrast to the above methods, Worstoff-DRO assignments are unrelated to the actual group labels. Despite being unrelated to the group labels, optimizing the DRO loss over these assignments also optimizes the GroupDRO objective as pointed out in the theoretical guarantees. We attribute this point as a strength in our paper.
>
> On the other hand, observe that all the methods JTT, GEORGE and EIIL are two-stage approaches in contrast to our proposed one-stage optimization. Thus, we cut the compute time by *half!*. Moreover, as discussed in Section 3.2 our method enjoys a direct guarantee and relation to the GroupDRO objective unlike the pseudo-labelling methods.
>
> ---
>
> **Question** While the theoretical results are correct, they are of limited impact. The gap between L_{GDRO} and L_{WDRO}, even the marginal distribution version, is likely to be large because the constraint set quite is large. In particular, for the points without group labels, only the marginal distributions have to approximately match, but given this the assignment of the individual group labels to the individual points is arbitrary, whereas in reality we could use the data features to better predict the probability of a given point having a particular group label. Thus, L_{WDRO} is a fairly pessimistic loss upper bound whose loss landscape may be quite different than that of L_{GDRO}; indeed, this is borne out by the empirical results.
>
> **Answer** Thanks for the insightful question! We note that the assignments are not completely arbitrary, they are a function of loss values of the samples. This is because the marginal constraint is applied alongside the Worstoff-DRO objective. The Example 3 in Appendix A.2 highlights this point and shows that the high loss samples are assigned to the worstoff groups (high $q/\bar p$). In contrast to the UnsupDRO baseline that only optimizes over the high-loss samples, observe that our approach upweights (via a large $q/\bar p$) the high loss samples and downweights the other samples.
>
> While it was tempting to use the group label predictions (from the data) as a regularizer to make the assignments closer to the ground-truth labels, unfortunately we haven’t found theoretical ground to use them safely and so we restricted ourselves as discussed in the previous answer.

---

> > ### Comment · Reviewer_Jwqz · 2021-11-25
> > **Thanks for the response**
> >
> > - I accept that in some situations, we may not want to estimate group labels for certain reasons. However, JTT just upweights the misclassified examples; it is difficult to imagine a setting in which such a simple method would be "harmful" (identifying misclassified examples always needs to be done even to just compute the accuracy of any model).
> >
> > - Moreover, even if group-label-estimation methods like GEORGE and EIIL could be undesirable in such settings, it would still be good to show comparisons where applicable, to understand how much performance is being sacrificed in order to maintain privacy.
> >
> > - It's not true that "optimizing the DRO loss over these assignments also optimizes the GroupDRO objective." The minimizers of the WDRO loss and GroupDRO loss are not necessarily the same.
> >
> > - It is true that JTT, GEORGE, and EIIL are two-stage approaches. However, the first stage of EIIL involves training a model for just one epoch, which is an insignificant amount of added compute (in Appendix E of the EIIL paper, they show that this only adds 0.4% compute time for Waterbirds, for example). So, it's not necessarily true that your method cuts compute time by half. Moreover, given the huge gap in min-group accuracy, it is likely that any of the aforementioned methods could be trained for half the time and still outperform Worst-Off DRO, so the compute argument is not compelling for this reason.
> >
> > - The authors should also cite and compare to CVaR DRO (https://arxiv.org/abs/2010.05893, NeurIPS '20), which does *not* attempt to estimate group labels (and does not require multiple "stages" in the same sense as JTT/GEORGE/EIIL - instead, they train an ERM model and then fine-tune the prediction head of that model). As shown in the JTT paper, CVaR DRO achieves 75.9% worst-group accuracy on Waterbirds and 64.4% on CelebA, both of which substantially exceed the performance of Worst-Off DRO (despite the fact that CVaR DRO does not require *any* group labels to be known during training, which seems to make it even more well suited to the proposed setting in which knowledge of group labels could be "misused").
> >
> > =====
> >
> > Question to the authors: How many validation set examples with group labels are used for model selection? It would be good to have a similar table to Table 1 except for the validation set.

---

> > > ### Author Response · Authors · 2021-11-28
> > > **Thanks for the comments!**
> > >
> > > We thank the reviewer the detailed comments and nice suggestions!
> > >
> > > ---
> > >
> > > **Question** It's not true that "optimizing the DRO loss over these assignments also optimizes the GroupDRO objective." The minimizers of the WDRO loss and GroupDRO loss are not necessarily the same:
> > >
> > > **Answer** We discuss below that the statement holds true because the Worstoff-DRO objective upper bounds the GroupDRO objective. We provide a short proof elaborating on the point. Notations are from the paper.
> > >
> > > $\text{GDRO}(w, q)  \le \text{WDRO}(w, q) \quad \forall w,q \quad -(1)$. From Lemma $1$ in the paper.
> > >
> > > Define, $q_W^* = \text{argmax}_q \text{WDRO}(w, q)$, $q_G^* = \text{argmax}_q \text{GDRO}(w, q)$
> > >
> > > From the above definitions, we have $\text{WDRO}(w, q) \le \text{WDRO}(w, q_W^*) \quad - (2)$, $\text{GDRO}(w, q) \le \text{GDRO}(w, q_G^*)$
> > >
> > > $\text{GDRO}(w, q_G^*) \le \text{WDRO}(w, q_G^*)$ (from $(1)$)
> > >
> > > $\text{WDRO}(w, q_G^*) \le \text{WDRO}(w, q_W^*)$ (from $(2)$)
> > >
> > > Hence, $\text{GDRO}(w, q_G^*) \le \text{WDRO}(w, q_W^*)$
> > >
> > > Therefore, minimizing RHS implies minimizing LHS.
> > >
> > > ---
> > >
> > > **Question** The authors should also cite and compare to CVaR DRO.
> > >
> > > **Answer** Good Suggestion! CVaR DRO, a coherent risk measure, optimizes over a certain $\alpha$-sized sub-populations within the training dataset. In effect, the method is similar to UnsupDRO where the size of the sub-population is controlled by a threshold $\eta$ on the loss value.
> > > As stated in the JTT paper (3.2.2 https://arxiv.org/pdf/2107.09044.pdf), the size of the selected sub-population ( determined via either the $\alpha$-parameter in CVaR DRO or the $\eta$-parameter in UnsupDRO) needs to be close to the size of the smallest group. This requires a wider hyper-parameter search space for $\alpha$ / $\eta$ parameters. Our experiments justify this need, Table 3 of Appendix A.1 shows that the search space of UnsupDRO is twice  relative to Worstoff-DRO to attain comparable accuracies. A wider search space contributes to a harder model selection procedure. Consequently,  scenarios where extensive search is not possible (eg, small validation set/dataset regimes) could result in incorrect/unstable model selection . On the other hand, CVaR-DRO / UnsupDRO train only on the highest loss samples while discarding the rest. In contrast, Worstoff-DRO does not discard any sample rather downweights/upweights as per the worst-off group assignment. This property aids in maintaining a high overall accuracy besides reaching good minority group accuracy.  Lastly, with regards to the direct comparison to CVaR DRO’s accuracy values from the JTT paper, kindly note the difference in the search spaces Table 10 of https://arxiv.org/pdf/2107.09044.pdf and Table 3 in the Appendix A.1 of our paper.
> > >
> > > ---
> > >
> > > **Question**  claim about reducing the computations by half JTT/EIIL/GEORGE
> > >
> > > **Answer** We concur with the reviewer’s opinion that a two-stage model may not always have twice the compute cost relative to a one-stage model. Algorithmic specific adjustments (such as single epoch training/parallelization) may reduce the compute time. A two-stage model, however, introduces a new set of hyper-parameters that need to be tuned such as the first stage epochs $T$ in JTT. The presence of the first stage also creates challenges where the errors from previous stages are propagated downstream or a failed first stage leads to an unsuccessful second stage. In case of JTT, the first stage model could fail due to the model overfitting to the training data as noted by the authors, whereas in EIIL/GEORGE inaccurate group inference may block invariant learning besides raising ethical issues as discussed before.
> > >
> > > ---
> > >
> > > **Question** How many validation set examples with group labels are used for model selection?
> > >
> > > **Answer** For CMNIST, $66$% of randomly selected samples are used for training, $17$% for validation and $17$% for testing.
> > >
> > > In the Adult dataset, the official train-test splits provided with the dataset were used. From the training dataset, $50$% of the samples were set aside for validation.
> > >
> > > For the CelebA dataset, the official train-val-test splits were used.
> > >
> > > Lastly, in Waterbirds, the train and test splits are from the CUB dataset. $20$% of the training data is set aside for validation, similar to GroupDRO paper.

---

### Decision · Program_Chairs · 2022-01-20

**Decision:**

Reject

**Comment:**

This paper introduces a novel method for learning distributional robust machine learning models when only partial group labels are available to improve performance of learning algorithms on minority groups.

Pros: The paper is well motivated and written.  The ideas are interesting.  Most work on distributional robust optimization (DRO) are in unsupervised settings where group information is not available.  They provide an approach for the semi-supervised setting through a constraint set.

Cons:
The empirical results do not show better performance over unsupervised baselines as pointed out by reviewers.

The authors claim one of the benefits of their proposed approach is a one-stage approach, in contrast to competing models that require a two-stage approach; hence, allowing their approach to reduce compute time.  It’ll be helpful to strengthen this point by showing time comparisons.

Missing labels in this case due to participants withholding sensitive information is not an MCAR case, but the proposed work makes an MCAR assumption.  It’ll help to add a discussion and point out such limitations of the approach.

Summary:  This paper has novel and interesting ideas, but still has several issues as pointed out by the reviewers before it is ready for publication.